# Comparison of Ultrasound Contrast between H_2_O_2_-Responsive Nanoparticles and Microbubble Contrast According to Muscle Injury in Rat Models

**DOI:** 10.3390/diagnostics13213320

**Published:** 2023-10-26

**Authors:** Da-Sol Kim, Nanhee Song, Dongwon Lee, Gi-Wook Kim

**Affiliations:** 1Department of Physical Medicine and Rehabilitation, Jeonbuk National University Medical School, Jeonju 54907, Republic of Korea; murunoon@naver.com; 2Research Institute of Clinical Medicine—Biomedical Research Institute, Jeonbuk National University Hospital, Jeonju 54907, Republic of Korea; 3Department of Bionanotechnology and Bioconvergence Engineering, Jeonbuk National University, Jeonju 54896, Republic of Korea; nanhee9710@gmail.com (N.S.); dlee@jbnu.ac.kr (D.L.); 4Department of Polymer Nano Science and Technology, Jeonbuk National University, Jeonju 54896, Republic of Korea

**Keywords:** ultrasonography, contrast agent, microbubbles, nanoparticles, musculoskeletal diseases

## Abstract

Ultrasound contrast agents are clinically used for diagnosis of internal organs, but ultrasound contrast agents are rarely applied clinically in musculoskeletal disorders. Our study aims to comparatively analyze the differences between ultrasonographic images through peri-injury injection of the clinically used microbubble and researched nanoparticle contrast agents in various muscular injury models. To compare contrast-enhanced images in different muscle injury models, we prepared groups of rats with sham, laceration, punch, contusion, and toxin injection injuries. We measured H_2_O_2_ levels using the Amplex Red assay by extracting tissue from the damaged area. As comparative contrast agents, SonoVue^®^, a commercially available microbubble contrast agent, and poly(vanillinoxalte) (PVO) nanoparticles, which are H_2_O_2_-responsive nanoparticles, were used. The difference in contrast between the two contrast agents was recorded as an ultrasound movie, and J-image software 1.53p was used to quantify and analyze the maximum and minimum echogenicity values of the images after contrast enhancement. In the Amplex red assay for the highest H_2_O_2_ level in each muscle injury model, the maximum level showed 24 h after the modeling. In the sham rats, PVO injection showed no increased echogenicity except at the needle insertion site, but SonoVue^®^ injection showed increased echo signal throughout the injected muscle immediately after injection. One day after the preparation of the lesion, PVO and SonoVue^®^ were injected into the lesion site and ultrasound was performed on the lesion site. After the injection of PVO nanoparticles, contrast enhancement was observed at the lesion site immediately. SonoVue^®^ injections, on the other hand, showed a widespread pattern of echo signals and an increase in echo retention only at the lesion site over time, but this was not clear. There were statistically significant differences between the highest and lowest echogenicity in PVO and SonoVue^®^ contrast-enhanced images in all models. Contrast enhancement lasted more than 3 h in the PVO injection, but disappeared within 3 h in the SonoVue^®^ injection. PVO nanoparticles showed the possibility of physiologic contrast by CO_2_ generated by conjugation with H_2_O_2_ generated by muscle injuries, and SonoVue^®^ injection observed the possibility of microbubble contrast as a contrast agent with a pooling effect that lasts longer on the lesion. Further research is needed to investigate the use of various ultrasound contrast agents, including nanoparticles, in musculoskeletal disorders, as well as the potential for further utilities of microbubble contrast agents.

## 1. Introduction

Diagnosis and guided treatment using ultrasonography imaging have the major advantages of posing no radiation risk, low cost, and permitting simultaneous diagnosis and intervention using real-time imaging. To increase the effectiveness of ultrasound imaging, studies on ultrasound contrast agents have been actively carried out. Commercially available ultrasound contrast agents use “microbubbles” and react with alternating pressure waves, which have great compressibility to expand and contract, resulting in an oscillatory motion. This mechanism is acknowledged to scatter ultrasonic energy in a superior manner than solids or liquids [1,2,3]. Microbubble ultrasound contrast agents have been commonly used for detecting malignant or benign lesions in the liver and kidney [4].

Studies on ultrasound contrast agents for use with musculoskeletal disorders have primarily used second-generation microbubble contrast agents, such as SonoVue^®^ (Bracco, Milan, Italy), Definity^®^ (Lantheus, MA, USA), and Sonazid^®^ (Daiichi Sankyo, Tokyo, Japan), through ultrasonography with blood retention contrast agents injected into the distal microcirculation. However, intravenously administered microbubble ultrasound contrast agents are limited in their ability to detect inflammatory diseases (such as rheumatoid disease), peripheral vascular disease, or musculoskeletal tumors, which are commonly accompanied by changes in vascular structure [4,5,6,7,8,9,10]. Further research is needed to investigate the detection of muscle and ligament or microscopic damage using ultrasound contrast agents and the proper injection method in musculoskeletal disorders.

Nanoparticle contrast agents have been studied as new options for ultrasound contrast agents. They have the advantages of small particle size, controllability, availability of biological reaction, and blood circulation stability [11]. However, in musculoskeletal disorders, the use of nanoparticle ultrasound contrast agents has been reported on in a limited manner, with studies focusing on skeletal muscle disorders including contusion injury, ischemic muscle injury, muscular dystrophy, and inflammatory arthritis [11,12,13,14,15,16,17,18]. Previous studies have mainly focused on contrast enhancement using the nanoparticle ultrasound contrast agents in a single model. Further studies are needed to investigate the characteristics, including echogenicity and time-course, as well as differences in multiple types, of musculoskeletal disorder models.

This study compared ultrasound-enhanced images of four types of rat models with musculoskeletal injuries using a microbubble ultrasound contrast medium (SonoVue^®^), and hydrogen peroxide-sensitive nanoparticles (PVO; poly(vanillinoxalte)) that a previous study reported on as having the potential of musculoskeletal ultrasound contrast agents in muscle injury models [19]. We hypothesized that enhancement using microbubble ultrasound contrast medium might be related to tissue defect of the wound via the pool effect, while enhancement using the nanoparticle contrast agent might be related to physiological damage and the production of hydrogen peroxide. Our study aims to comparatively analyze the differences ultrasonographic images through peri-injury injection of the nanoparticle and microbubble contrast agents in various muscular injury models.

## 2. Materials and Methods

### 2.1. Preparation of PVO Nanoparticles and SonoVue^®^

Consistent with previous reports, PVO was produced as a result of a reaction between oxalyl chloride and an acid-cleavable vanillin derivative [20]. Oxalyl chloride and the acid-cleavable vanillin derivative (3.941 mmol) were mixed in a flask of 25 mL dried dichloromethane and pyridine (9.8 mmol). The flask was placed in an ice bath and oxalyl chloride was added (3.941 mmol). The reaction continued for 6 h at room temperature. PVO was obtained using dichloromethane/water and precipitated in cold hexane. The chemical structure of PVO was verified by nuclear magnetic resonance spectroscopy after purification. PVO nanoparticles were obtained by sonification of 50 mg of PVO dissolving in dichloromethane and 5 mL of 5% poly(vinyl alcohol) (PVA), homogenization was conducted with a fine oil/water and PVA emulsion, evaporation of solvent, and centrifuge. PVO nanoparticles responded to peroxide and released vanillin and carbon dioxide gas (Figure 1A).

SonoVue^®^ consists of sulfur hexafluoride (SF_6_) gas and phospholipidic monolayer shells (Figure 1B). Using the MiniSpike transfer system, 25 mg of dry, lyophilized powder Macrogol 4000, Distearoylphosphatidylcholine, Dipalmitoylphosphatidylglycerol Sodium, Palmitic acid in an atmosphere of SF_6_ and 5 mL of sodium chloride (9 mg/mL, 0.9%) solution were transferred into the syringe, which was completely shaken to avoid separation of the liquid and solid parts of the lyophilisate.

### 2.2. Four Types of Muscle Injury Models in Rats

Sprague–Dawley rats (8-week-old males, Orient Bio, Seongnam, Republic of Korea) were anesthetized by intraperitoneal administration of the mixture of ketamine and xylazine (8:1). After anesthetization, isoflurane (Ifran Solution, Hana Pharm, Seoul, Republic of Korea) with 100% oxygen was delivered to the rats using a vaporizer for maintenance of anesthetization. All muscle injuries were induced to the middle of the knee joint and the Achilles tendon. Animal experiments were approved by the Institutional Animal Care and Use Committee (IACUC) of Jeonbuk National University Hospital (CUH-IACUC-2021-34) and conducted in accordance with relevant regulations and guidelines. The sixteen rats were randomly allocated to four muscle injury groups (*N* = 4 per group): laceration, punch, contusion, and toxin injection injury groups. Additionally, we subdivided two groups via the depth of muscle injury: the inner layer group (laceration and punch groups) and the outer layer group (contusion and toxin injection groups).

#### 2.2.1. Laceration Injury

The rats were placed in the lateral decubitus position, with the ankle slightly dorsiflexed. After hair clipping, depilation, and skin cleaning, incisions 1–1.5 cm in length and 5–10 mm in depth were made with a surgical blade parallel to the tibia and induced to the middle of the knee joint and Achilles tendon, through the subcutaneous tissue down to the triceps surae muscles (Figure 2A). The skin was then sutured with 4.0 suture string.

#### 2.2.2. Punch Injury

The rats were placed in a lateral decubitus position. After skin preparation (hair clipping, depilation, and skin cleaning), 1–1.5 cm skin incisions were made and round muscle injuries were made by a 1.5 mm biopsy punch tool with less than 1.5 mm in depth, through the subcutaneous tissue down to the triceps surae and plantaris muscles (Figure 2B). The injury was induced in the middle of the knee joint and Achilles tendon.

#### 2.2.3. Contusion Injury

The rats were placed in the lateral decubitus position, with the knee extended and the ankle dorsiflexed. A compression instrument (TMS-PRO^®^, FTC Corp., Sterling, VA, USA) was used to induce a contusion injury in the triceps surae and plantaris muscles at the middle of the knee joint and Achilles tendon. The muscles were pressed by a round flat probe (160-N force) descending at a rate of 20 mm/min until the thickness reached 3 mm (Figure 2C).

#### 2.2.4. Toxin Injection Injury

The rats were placed in the lateral decubitus position, with the knee extended and the ankle slightly dorsiflexed. To produce a toxin injury model, 2 µL carrageenan (10 mg/mL) was injected with a 25 Gauge syringe into the gastrocnemius muscles in the middle of the knee joint and Achilles tendon (Figure 2D) [21].

### 2.3. Determination of the Level H_2_O_2_ in the Site of Muscle Injuries

A CO_2_ bubble is formed as a result of the reaction between PVO and H_2_O_2_ at the injured site (Figure 1A). To optimize the ultrasound contrast enhancement of PVO, we measured the level H_2_O_2_ at the site of the muscle injuries. The tissues of the triceps surae muscles were excised 6 h after the injury and lysed in PBS at a concentration of 10 mg/mL using a homogenizer. Tissue lysates were centrifuged at 8000× *g* for 10 min and the supernatants were carefully taken. The level of H_2_O_2_ in the supernatant was determined using the Amplex red assay kit (Invitrogen, Carlsbad, CA, USA) and a microplate reader (Biotek Instruments, Winooski, VT, USA). The fluorescent intensity (FI) of the injured tissues was compared with those of non-injured tissues.

### 2.4. Ultrasonographic Imaging

An ultrasonographic imaging instrument (Zone Ultra, Zonare Medical Systems, San Francisco, CA, USA) with a linear transducer at 5 to 14 MHz was used for image acquisition. The probe’s dimensions were 62 mm × 10 mm with a viewing width of 55 mm. The ultrasonographic images were obtained from the long and short axis view by a physician with an extensive experience in musculoskeletal ultrasonography. Contrast-enhanced ultrasonography was performed using SonoVue^®^ and PVO nanoparticles, and the images were obtained 1 day after the muscle injuries and until the contrast enhancement disappeared. The contrast agents were injected perilesional site. Echogenicity was quantified using Image J, with the musculoskeletal ultrasound professional manually selecting the region of interest (ROI) with a size of 0.5 × 0.5 cm in the longitudinal view of the ultrasound images. The highest and lowest echogenicity values for the muscle injury site were compared.

### 2.5. Statistical Analysis

All statistical analyses were conducted using SPSS version 23.0. The Mann–Whitney U test were performed on the echo generation values of the upper and lower limit ROIs of *n* = 4 for each contrast agent. A *p*-value of less than 0.05 (95% confidence level) was considered statistically significant.

## 3. Results

### 3.1. Determination of H_2_O_2_ Level in Four Injury Models by Amplex Red Assay

An Amplex red assay was conducted at three time points (3 h and 24 h) after the induction of muscular injuries in the four models. The results showed that the H_2_O_2_ levels were highest in all four models 24 h after the injury (Figure 2A–D). According to peak H_2_O_2_ level, the injection of contrast agents and the measurement of contrast-enhanced images were performed one day after the injury in all four models.

### 3.2. Contrast Enhanced Ultrasound Image Using PVO and SonoVue^®^ in Sham Group

First, ultrasound images were compared after intramuscular injection of PVO and SonoVue^®^ into the triceps surae muscle in sham rats. In the PVO injection, there was no difference in echogenicity on the ultrasound except at the needle insertion site. In contrast, SonoVue^®^ injection showed increased echo signals throughout the muscle compartment immediately after injection and disappeared within 10 min (Figure 3).

### 3.3. Ultrasound Images after PVO and SonoVue^®^ Injection in the Muscle Injured Models

After directly injecting PVO and SonoVue^®^ into the muscles around the lesion, ultrasound was performed on the damaged area. After injection of PVO nanoparticles, enhanced ultrasonographic contrast was observed within 30 to 60 min for 1 to 2 s at the lesion sites. On the other hand, a widespread pattern of highly intense echo signals in SonoVue^®^ was seen around the injection site, and increased echo retention was seen only at the lesion site over time, but was unclear.

#### 3.3.1. Laceration Model

The contrast-enhanced ultrasound image using PVO nanoparticles and SonoVue^®^ exhibited linear-shaped enhancement in both the longitudinal and short view (Figure 4A). In the PVO group, the peak echogenicity was observed 30 min after injection and remained constant over a period of more than 3 h. On the other hand, SonoVue^®^ showed peak echogenicity 3 min after injection, which slowly decreased the echogenecity (Figure 5A). Statistical analysis showed significant differences between the highest and lowest echogenicity in both PVO and SonoVue^®^ groups (*p* = 0.001 and *p* = 0.029, respectively) (Figure 6A).

#### 3.3.2. Punch Models

In the contrast enhanced ultrasound image taken with PVO nanoparticles, donut shape enhancement in the longitudinal and dot shape in the short view was seen, and peak echogenicity was observed 30 min after injection, which then gradually decreased over the course of 3 h (Figure 4B and Figure 5B). In the SonoVue^®^, an area with enhanced contrast was seen on the delayed image, not clear. The peak echogenicity was observed immediately after injection of the contrast media, which also gradually declined over 3 h. There were statistically significant differences between the highest and lowest echogenicity in both PVO and SonoVue^®^ contrast-enhanced images (*p* = 0.029 in both PVO and SonoVue^®^, Figure 6B).

#### 3.3.3. Contusion Models

In the PVO contrast-enhanced image, the enhancement was observed in the middle layer of the injured muscle, and peak echogenicity was seen 30 min after injection. This gradually decreased over the course of 3 h (Figure 4C and Figure 5C). The SonoVue^®^ also maintained contrast enhancement in the mid-muscle region on delayed imaging, and peak echogenicity was seen 10 min after contrast, disappearing over 3 h (Figure 4C and Figure 5C). There were statistically significant differences between the highest and lowest echogenicity in PVO and SonoVue^®^ contrast-enhanced images (*p* = 0.000 in PVO and *p* = 0.029 in SonoVue^®^, see Figure 6C).

#### 3.3.4. Toxin Injection Models

In the contrast enhancement image, PVO was observed within the gastrocnemius muscle immediately after injection. And, SonoVue^®^ was observed on delayed imaging, but was not clear. The peak echogenicity of PVO was observed 10 min after injection, after which it gradually decreased over the course of 3 h (Figure 4D and Figure 5D). The peak echogenicity of SonoVue^®^ was observed 10 min after contrast, disappearing over 3 h. There were statistically significant differences between the highest and lowest echogenicity in both PVO and SonoVue^®^ contrast-enhanced images (*p* = 0.029 in both PVO and SonoVue^®^, Figure 6D).

## 4. Discussion

The microbubble ultrasound contrast medium consists of a gas interior with a shell surrounding a structure of proteins, lipids, and polymers [3]. Each structure is 2–10 µm smaller than red blood cells and passes through capillaries via intravenous injection to reached target organs through pulmonary circulation and systemic circulation. Commercialized contrast agents on the market today are generally second-generation, meaning they are more stable in the human circulatory system and use (1) gas with low solubility, (2) diversified shell materials, and (3) are of higher molecular weight, compared to first-generation microbubble contrast agents [22,23,24]. Representative examples include SonoVue^®^, Definity^®^, Sonazid^®^ with a phospholipid shell, and Optison^®^ with an albumin shell [25].

Nanoparticle ultrasound contrast agents have been studied as a means of overcoming the limitations associated with microbubble contrast agents. For example, the short circulation time of microbubble contrast agents results in difficulties in imaging examinations, and the relatively large size particle decreases effective extravasation and absorption in the tumor tissue, resulting in less efficient contrast enhancement. Particles with a size of 200–700 nm can pass through the capillary wall accompanied by tumors and inflammation, while particles with a size of 10–60 nm can be absorbed into the lesion [26]. Nanoparticle ultrasound contrast agents can be divided into two types according to their structure; (1) a gas or liquid-based nanoparticle, surrounded by a shell, such as lysosomes, polymers, micelles, dendrimers, emulsions, quantum dots, silica, or polymer nanoparticles, with an internal gas (e.g., nitrogen, carbon dioxide, perfluorocarbon, sulfur hexafluoride) or liquid (similar to microbubble ultrasound contrast agent) and (2) a solid-based nanoparticle with a high scattering acoustic signal structure and signal [27,28,29,30,31,32]. Our previous studies reported the potential of a musculoskeletal ultrasound contrast agent using liquid-based hydrogen peroxide-sensitive nanoparticles (PVAX; poly(vanillyl alcohol-co-oxalate) and PVO; poly(vanillinoxalte)) in a muscle contusion model [19,33].

This study comparatively analyzed the differences between ultrasonographic images through peri-injury injection of the nanoparticle (PVO) and microbubble (SonoVue^®^) contrast agents in various muscular injury models. There has been no study comparing the characteristics of ultrasound imaging using microbubble and nanoparticle contrast agents in musculoskeletal ultrasound. In the Amplex red assay for the highest H_2_O_2_ level in each muscle injury model, the maximum level showed 24 h after the modeling, and there were no significant differences among the four groups. In the sham rats, PVO injection showed no increased echogenicity except at the needle insertion site, but SonoVue^®^ injection showed increased echo signal throughout the injected muscle immediately after injection. One day after the preparation of the lesion, PVO and SonoVue^®^ were injected into the lesion site and ultrasound was performed on the lesion site. After the injection of PVO nanoparticles, contrast enhancement was observed at the lesion site immediately. SonoVue^®^ injections, on the other hand, showed a widespread pattern of echo signals and an increase in echo retention only at the lesion site over time, but this was not clear. The difference between the two contrast agents is that, in SonoVue^®^, the microbubble contrast was maintained longer in the anatomical damaged muscle area, which is a pooling effect, whereas in PVO, it is presumed to be a physiological effect because the increased contrast caused by CO_2_ generated by the conjugation of PVO and H_2_O_2_ generated during muscle damage.

Although the highest signal intensity was similar between PVO and SonoVue^®^, the echogenicity in PVO exhibited longer persistence than SonoVue^®^. In all four muscle injured models, there were statistically significant differences between the highest and lowest echogenicity in both PVO and SonoVue^®^ contrast-enhanced images. With respect to contrast enhancement time, peak enhancement was observed 30 min after contrast in the PVO. In the case of SonoVue^®^, peak enhancement was observed within 10 min after injection and withdrawal of the contrast agent. Although contrast enhancement was maintained over 3 h in both the PVO and SonoVue^®^ contrast-enhancement ultrasound images, the enhancement decreased much more slowly in the PVO group than the SonoVue^®^ group. There were no significant differences in the peak echogenicity between PVO and SonoVue^®^ contrast-enhanced images in all models. PVO nanoparticles enhance contrast after a CO_2_ bubble is generated through a chemical reaction between H_2_O_2_ and PVO in the damaged area, while contrast enhancement using SonoVue^®^ increases blood vessel formation and anatomical deprivation in the damaged area. These contrast mechanisms might be related to the delayed enhancement of PVO nanoparticles, compared to SonoVue^®^. The contrast enhancement lasted more than 3 h longer in the PVO group than the SonoVue^®^ group, which suggests that contrast enhancement is maintained as long as CO_2_ bubble generation occurs. The more prolonged enhancement could offer increased diagnostic value for lesions—illustrated, for instance, by time-lapse ultrasound images across acute, subacute, and delayed phases. Also, the prolonged enhancement itself would reduce the diagnostic constraints posed by brief contrast durations. Also, our prior study, which investigated the diagnostic and therapeutic impacts of PVO nanoparticles in a rat contusion injury model, pointed towards the antioxidant and anti-inflammatory properties of PVO nanoparticles [19]. The sustained presence of PVO nanoparticles might suggest a heightened therapeutic efficacy in attenuating primary injury and subsequent inflammation at the lesion site. However, translational challenges associated with this prolonged enhancement, such as potential delays in excretion and related toxicity, need attention. Further research is required.

Contrast enhancement using perilesional SonoVue^®^ was observed over or within 3 h, which is much longer than that using intravenous SonoVue^®^ injection. The perilesional method could improve the short enhancement time of intravenous SonoVue^®^. Previous studies on contrast-enhanced ultrasound in musculoskeletal disorders via the IV route have highlighted diagnostic limitations stemming from short duration and low echogenicity, as well as efficacy issues in less vascularized lesions [4,5,6,7,8,9]. These challenges are likely linked to the intravenous permeability and stability of the ultrasound contrast agent. To address these limitations and enhance the effectiveness of the ultrasound contrast, we opted for peri-lesional injection. Moreover, most therapeutic interventions using ultrasound in musculoskeletal disorders perform peri-lesional or direct injections. The diagnostic and therapeutic application via perilesional injection seems to be efficient.

In this study, in order to minimize the subjective interpretation of the image of the ultrasound contrast agent, a Appendix A was added and the difference in contrast intensity and duration were compared through the J image. To our knowledge, there were no studies that compared differences in ultrasound images between two ultrasound contrast agents (nanoparticles and microbubble ultrasound contrast) after peri-lesion injection in rat models with various musculoskeletal disorders. PVO nanoparticles showed the possibility of physiologic contrast by CO_2_ generated by conjugation with H_2_O_2_ generated by muscle injuries, and SonoVue^®^ injection observed the possibility of microbubble contrast as a contrast agent with a pooling effect that lasts longer on the lesion. This author seems to believe in the higher possibility of nanoparticle contrast using physiologic effect than pooling effect for clinical use as a musculoskeletal ultrasound contrast agent. None of the sixteen rats in our study exhibited adverse events following treatment with PVO nanoparticles. However, the long-term effects were not assessed, indicating a need for further research.

## 5. Conclusions

PVO nanoparticles showed the possibility of physiologic contrast by CO_2_ generated by conjugation with H_2_O_2_ generated by muscle injuries, and SonoVue^®^ injection observed the possibility of microbubble contrast as a contrast agent with a pooling effect that lasts longer on the lesion. Further research is needed to investigate the use of various ultrasound contrast agents, including nanoparticles, in musculoskeletal disorders, as well as the potential for further utilities of microbubble contrast agents.

## Figures and Tables

**Figure 1 diagnostics-13-03320-f001:**
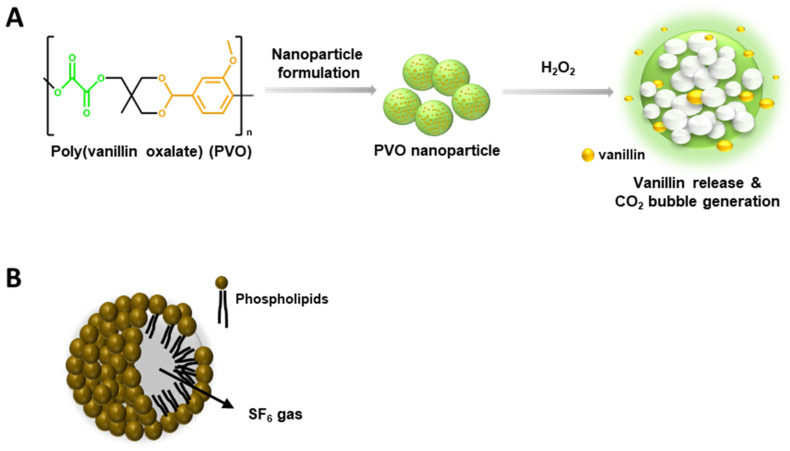
Characteristics of Poly(vanillinoxalate) (PVO) and SonoVue^®^. (**A**) A schematic diagram of H_2_O_2_-triggered CO_2_ bubble-generation by PVO nanoparticles. (**B**) Chemical structure of SonoVue^®^, a microbubble contrast agent.

**Figure 2 diagnostics-13-03320-f002:**
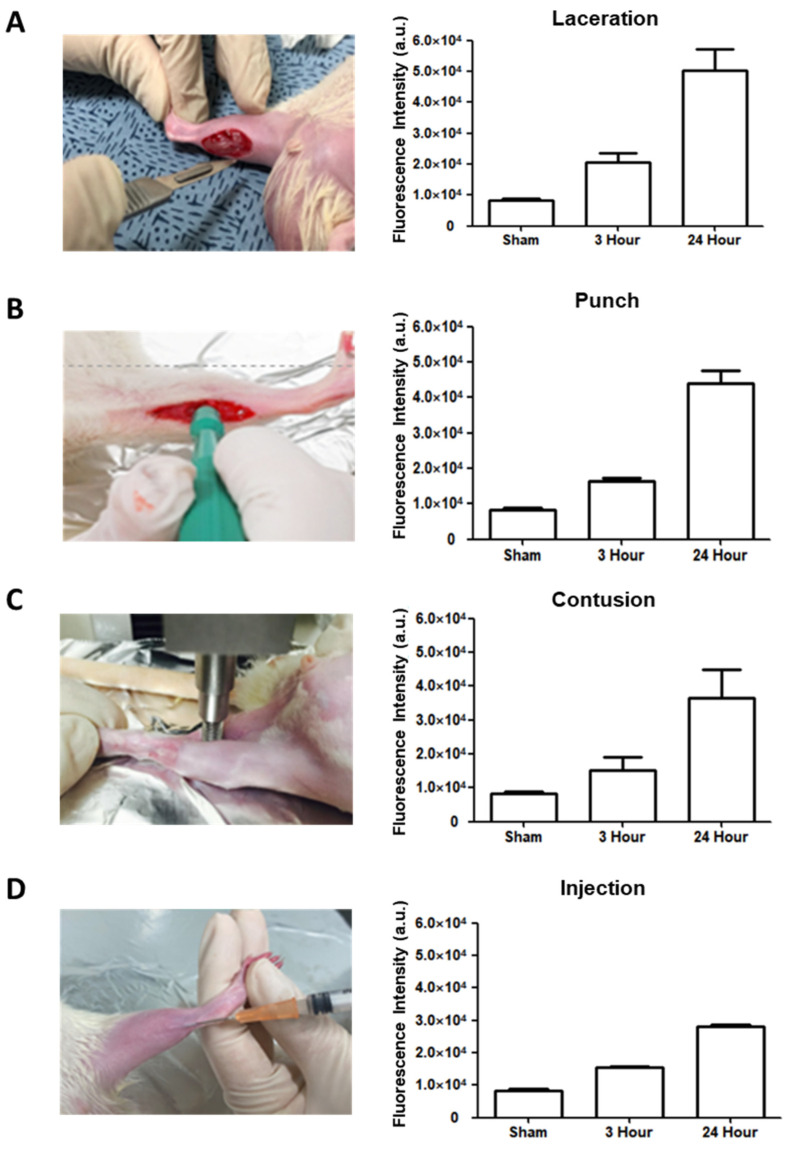
Rat modeling with four musculoskeletal injuries and determination of the level of H_2_O_2_ in each rat model by Amplex red assay. (**A**) Laceration model. (**B**) Punch model. (**C**) Contusion model. (**D**) Injection model.

**Figure 3 diagnostics-13-03320-f003:**
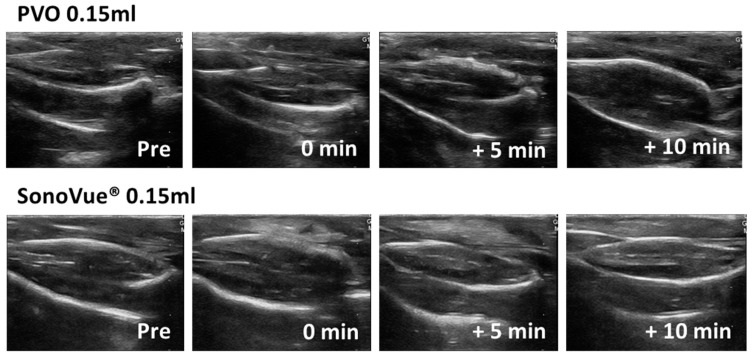
Ultrasound image after PVO and SonoVue^®^ injection in sham group.

**Figure 4 diagnostics-13-03320-f004:**
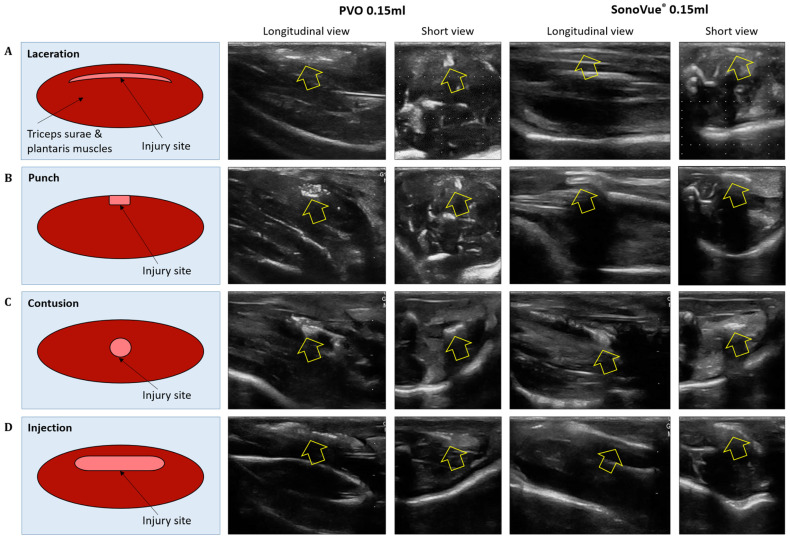
Ultrasound image of peak contrast enhancement in multiple musculoskeletal models after injection of PVO and SonoVue^®^. (**A**) Laceration model. (**B**) Punch model. (**C**) Contusion model. (**D**) Injection model.

**Figure 5 diagnostics-13-03320-f005:**
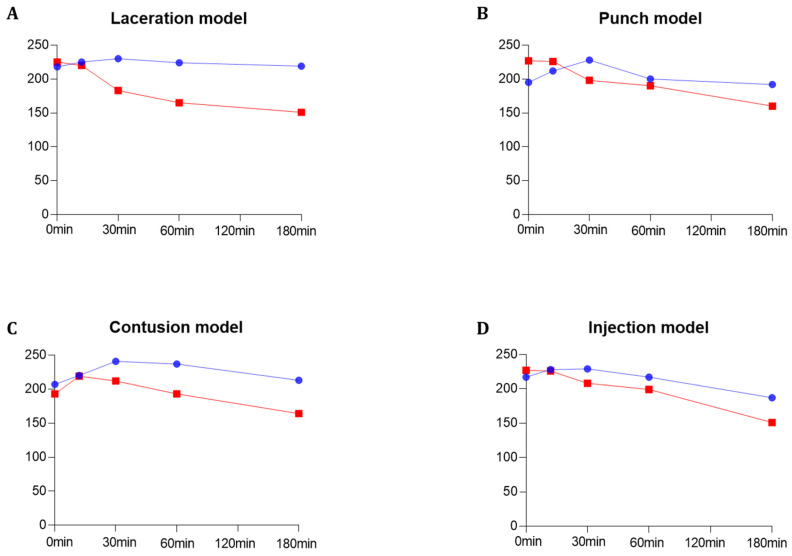
Quantitative grayscale analysis of the maximum echogenicity for PVO and SonoVue^®^ within the four multiple musculoskeletal rat models (laceration, punch, contusion, and injection models) using image J quantification software 1.53p (https://imagej.nih.gov/ij/ accessed on 4 March 2022). (**A**) Laceration model. (**B**) Punch model. (**C**) Contusion model. (**D**) Injection model. The blue graph represents PVO, while the red graph indicates SonoVue^®^.

**Figure 6 diagnostics-13-03320-f006:**
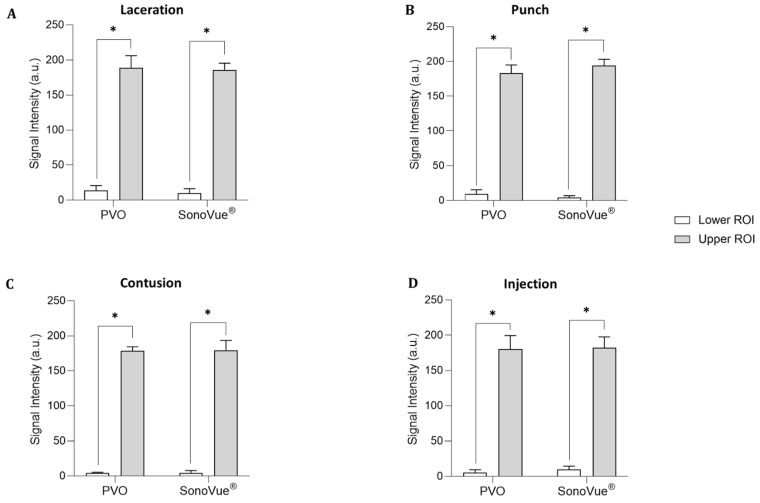
Quantitative grayscale analysis of the lower and upper ROI of contrast enhancement for PVO and SonoVue^®^ within the four musculoskeletal rat models (*N* = 4 per a model) using image J quantification software. (**A**) Laceration model. (**B**) Punch model. (**C**) Contusion model. (**D**) Injection model. An asterisk (*) indicates *p* < 0.05.

## Data Availability

The data used to support the findings of this study are available from the corresponding author upon request.

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
