# Peer review of "Comparison of Ultrasound Contrast between H2O2-Responsive Nanoparticles and Microbubble Contrast According to Muscle Injury in Rat Models"

_diagnostics, 2023, doi:10.3390/diagnostics13213320_

Round 1

Reviewer 1 Report

The authors report on the use of a new H2O2-responsive nanoparticle for contrast enhanced ultrasounds applications and compare it to a commercially available microbubble contrast agent.

In my opinion, this contribution doesn’t deserve publication in Diagnostics if not after a deep and substantial revision. First of all, the manuscript is very roughly written and several stylistic and substantial errors occur along all the text. Second, it seems that some of the results are misinterpreted, leading to incorrect conclusions. For example, the comments on page 7 about Figure 5 (reporting the most important results of the work) are not always accurate.

 Some major points to be addressed are:

1)     Pag 7 line 219-221: comments to figure 5A, which read as: “in the PVO group the peak echogenicity was observed 30 minutes after injection and gradually dissipated over a period of more 3 hours. On the other hand, SonoVue showed peak echogenicity 3 minutes after injection, which disappeared within 3 hours” are completely misleading: i) in the PVO group there isn’t a peak at 30 min; ii) the signal doesn’t dissipate gradually but remains constant; iii) for SonoVue the signal doesn’t disappear after 3 hours …it  remains only slightly lower than that of PVO.

2)     Pag 7 lines 228-234: all comments to Fig. 5B  are completely false …the author cannot underline differences which do not exist …the two profiles are very close one to the other

3)     General comments are reported in the discussion (pag  10 lines 322-325) which apply only for the contusion model , which is the same injury model already reported in the author’s previous paper  (ref 19) on the same PVO system, not for all models. Moreover, the authors write that for PVO contrast enhancement is maintained for 6-7 hours but no experimental evidence for this is showed in any of the figures.

Minor points:

1)     Pag 3, lines 119-120: the number of total used rats should be 16 not 14.

2)     Pag 6, lines 191-193: a paragraph from the template is left which has to be removed

3)     Pag 6, lines 210-213: Is this comment referred to SonoVue? The subject is missing

4)     Pag 7, lines 215-218: this sentence is almost incomprehensible; the English should be improved

5)     Èag 9, lines 279-282: This paragraph seems to have nothing to do with the rest of the text. What the authors meant?

Some paragraphs are almost incomprehensible. Extensive editing of English language is required

Reviewer 2 Report

Dear Authors,

I have thoroughly reviewed your paper, "Comparative Analysis of Ultrasound Contrast Agents in Musculoskeletal Disorders," and I must commend you for the well-executed study. Your research provides a substantial dataset, clear tabular representations, and effectively addresses the concept validation aspect. However, I do have a few questions regarding the clinical translational value, and I would appreciate your insights:

Toxicity and Biocompatibility:

The study provides preliminary insights into the toxicity and biocompatibility of PVO nanoparticles. However, a more comprehensive assessment of long-term effects and potential adverse reactions in vivo is needed before considering clinical applications.

IV Injection Feasibility:

The paper raises questions about the feasibility of IV injection for relatively larger PVO nanoparticles. Further investigation into optimal administration routes and addressing size-related challenges is essential for practical clinical use.

Clinical Translational Value:

The research highlights the contrast enhancement differences between SonoVue® and PVO nanoparticles, with PVO showing longer-lasting enhancement. However, it should also address translational hurdles and strategies for clinical implementation, enhancing its practicality.

Round 2

Reviewer 1 Report

The authors have addressed most of my requests/observations. Still two revisions have to be made, in my opinion, for the final publication of the paper.

1) PVO Data reported in Figure 5B are different from those included in the same figure of the "authors response file". The comments to data at pag 7 (3.3.2. Punch models) in the revised manuscript, better fit to those of the "authors response file". Please align Fig. 5B with real experimental data. 

2) Please remove paragraph at pag 9 lines 280-283 because it is out of place in the described context.

English was improved even if moderate editing still shoud be carried out.
